# The Beneficial Effect of the SGLT2 Inhibitor Dapagliflozin in Alleviating Acute Myocardial Infarction-Induced Cardiomyocyte Injury by Increasing the Sirtuin Family SIRT1/SIRT3 and Cascade Signaling

**DOI:** 10.3390/ijms25158541

**Published:** 2024-08-05

**Authors:** Yi-Hsiung Lin, Wei-Chung Tsai, Chien-Chih Chiu, Nai-Yu Chi, Yi-Hsueh Liu, Tien-Chi Huang, Wei-Tsung Wu, Tsung-Hsien Lin, Wen-Ter Lai, Sheng-Hsiung Sheu, Po-Chao Hsu

**Affiliations:** 1Division of Cardiology, Department of Internal Medicine, Kaohsiung Medical University Hospital, Kaohsiung Medical University, Kaohsiung 80756, Taiwan; caminolin@gmail.com (Y.-H.L.); azygo91@gmail.com (W.-C.T.); marchchi@gmail.com (N.-Y.C.); liuboy17@gmail.com (Y.-H.L.); 990314kmuh@gmail.com (T.-C.H.); 990322kmuh@gmail.com (W.-T.W.); lth@kmu.edu.tw (T.-H.L.); wtlai@kmu.edu.tw (W.-T.L.); sheush@cc.kmu.edu.tw (S.-H.S.); 2Center for Lipid Biosciences, Department of Medical Research, Kaohsiung Medical University Hospital, Kaohsiung Medical University, Kaohsiung 80756, Taiwan; 3Regenerative Medicine and Cell Therapy Research Center, Kaohsiung Medical University, Kaohsiung 80756, Taiwan; 4Graduate Institute of Clinical Medicine, College of Medicine, Kaohsiung Medical University, Kaohsiung 80756, Taiwan; 5Department of Internal Medicine, Faculty of Medicine, School of Medicine, Kaohsiung Medical University, Kaohsiung 80756, Taiwan; 6Department of Biotechnology, Kaohsiung Medical University, Kaohsiung 80756, Taiwan; cchiu@kmu.edu.tw; 7Department of Medical Research, Kaohsiung Medical University Hospital, Kaohsiung 80756, Taiwan; 8The Graduate Institute of Medicine, Kaohsiung Medical University, Kaohsiung 80756, Taiwan

**Keywords:** SGLT2 inhibitor, dapagliflozin, AMI, SIRTs, hypoxia

## Abstract

Sodium-glucose cotransporter-2 inhibitors (SGLT2i) have a variety of cardiovascular and renoprotective effects and have been developed as novel agents for the treatment of heart failure. However, the beneficial mechanisms of SGLT2i on cardiac tissue need to be investigated further. In this study, we established a mouse model of acute myocardial infarction (AMI) using coronary artery constriction surgery and investigated the role of dapagliflozin (DAPA) in protecting cardiomyocytes from hypoxic injury induced by AMI. In vitro experiments were done using hypoxic cultured H9c2 ventricular cells to verify this potential mechanism. Expression of the SIRT family and related genes and proteins was verified by qPCR, Western blotting and immunofluorescence staining, and the intrinsic potential mechanism of cardiomyocyte death due to AMI and hypoxia was comprehensively investigated by RNA sequencing. The RNA sequencing results of cardiomyocytes from AMI mice showed that the SIRT family may be mainly involved in the mechanisms of hypoxia-induced cardiomyocyte death. In vitro hypoxia-induced ventricular cells showed the role of dapagliflozin in conferring resistance to hypoxic injury in cardiomyocytes. It showed that SIRT1/3/6 were downregulated in H9c2 cells in a hypoxic environment, and the addition of dapagliflozin significantly increased the gene and protein expression of SIRT1, 3 and 6. We then verified the underlying mechanisms induced by dapagliflozin in hypoxic cardiomyocytes using RNA-seq, and found that dapagliflozin upregulated the hypoxia-induced gene downregulation, which includes *ESRRA*, *EPAS1*, *AGTRAP*, etc., that associated with SIRTs-related and apoptosis-related signaling to prevent H9c2 cell death. This study provides laboratory data for SGLT2i dapagliflozin treatment of AMI and confirms that dapagliflozin can be used to treat hypoxia-induced cellular necrosis in cardiomyocytes, in which SIRT1 and SIRT3 may play an important role. This opens up further opportunities for SGLT2i in the treatment of heart disease.

## 1. Introduction

Sodium-glucose cotransporter-2 inhibitor (SGLT2i) is called “the statins of the 21st century” because of its multiple cardiovascular (CV) and renal protection effect [1]. This drug was initially developed as an oral anti-diabetic drug for diabetes control. However, major CV outcome trials showed that SGLT2i can protect renal function and reduce hospitalizations of heart failure (HF) in diabetic patients [2,3,4]. Therefore, further randomized trials were conducted to evaluate the effect of SGLT2i in patients with HF or chronic kidney disease, and these studies also revealed significant benefits of SGLT2i in these two patient groups [5,6]. There are several possible mechanisms why SGLT2i has so many pleiotropic effects. Firstly, SGLT2i inhibits the sodium–hydrogen exchanger (NHE), which plays a critical role in cardiac cellular homeostasis. Inhibition of NHE by SGLT2i helps reduce intracellular sodium and calcium overload, thereby protecting the myocardium from ischemic injury [7]. Secondly, SGLT2i induces a metabolic shift in cardiac fuel utilization. This shift moves away from the energy-inefficient glucose metabolism towards the more efficient use of fatty acids and ketone bodies. This metabolic reprogramming enhances myocardial energetics and efficiency, contributing to improved cardiac function [8]. Thirdly, HF is often associated with functional iron deficiency, characterized by increased plasma hepcidin levels and lower iron content in tissues, which worsens mitochondrial function. SGLT2i has been shown to improve iron metabolism by reducing hepcidin levels, thereby increasing iron content in tissues and enhancing mitochondrial function. This mechanism further supports the cardioprotective effects of SGLT2i [9]. However, there are still a lot of mysteries and unknown mechanisms we do not understand for this drug.

Acute myocardial infarction (AMI) is one of the most important CV emergencies because of coronary artery stenosis or occlusion that further leads to hypoxia and ischemia in cardiac muscle tissue. Without adequate treatment, mortality is extremely high for this terrible disease. Percutaneous coronary intervention and optimal medical treatment can help physicians acquire better CV outcomes for patients of AMI [10,11,12]. However, AMI may further lead to several CV comorbidities such as HF, cardiac arrhythmia, and so on. According to previous HF studies [5,6,13,14,15], SGLT2i can improve not only CV outcomes, but also quality of life with HF, and it might be a novel treatment for patients with AMI.

Recent studies have also highlighted the benefits of SGLT2i in preclinical and clinical settings beyond HF. A study using a porcine model of left anterior descending coronary artery occlusion found that SGLT2i ameliorates myocardial ischemia–reperfusion injury and significantly reduces myocardial infarction (MI) size [16]. Additionally, clinical data from type 2 diabetic patients indicate that those treated with SGLT2i have smaller MI sizes, as evidenced by lower peak troponin levels compared to those of non-SGLT2i users [17]. There are also three important clinical trials of SGLT2i for AMI patients, including EMMY, EMPACT-MI and DAPA-MI studies [18,19,20]. In the EMMY study, empagliflozin was associated with a significantly greater NT-proBNP reduction, accompanied by a significant improvement in echocardiographic functional and structural parameters [14]. In the DAPA-MI study, although the addition of dapagliflozin did not reduce the composite endpoint of CV death or heart failure hospitalization, it significantly improved cardiometabolic outcomes even under low event rates in the trial [19]. The EMPACT-MI study also showed that empagliflozin initiated within 14 days after MI did not decrease the composite outcome of CV death or HF hospitalization, but it could reduce the risk of HF after a heart attack [21]. These three studies still suggest that SGLT2i may have benefits for AMI patients, although lots of mechanisms are still unknown and need further evaluation.

Sirtuins may play a role in many cellular processes, including apoptosis, DNA repair and lipid/glucose metabolism [22,23]. SIRT1 is the most conserved NAD^+^-dependent protein deacetylase and has emerged as a regulator of glucose metabolism in mammals. For gluconeogenesis, SIRT1 plays dual and complex roles in the short-term and prolong phases [24,25]. Studies have shown that certain types of deacetylase may have protective effects against endothelial dysfunction, atherosclerosis, cardiac hypertrophy and reperfusion injury [26,27,28]. However, there are few experimental data regarding deacetylase in patients with AMI. Clinically, based on these preliminary data and a recent active clinical study (the EMMY study), there must be some yet unrevealed beneficial mechanism for SGLT2i in the treatment of AMI.

The latest study by Chen et al. indicates that the levels of SIRT1, NF-κB and CD40L in the plasma of AMI patients are significantly increased, and concluded that combined use with TNT can improve the diagnostic accuracy of AMI [29]. However, the expression and detailed roles of the SIRT family proteins in cardiac tissue have not yet been fully discussed. SIRT1 and other SIRTs have been shown to play important roles in regulating hypoxic stress-associated mechanisms, including AMPK, TP53 and many other genes that are possibly regulated by dapagliflozin. Therefore, the role of dapagliflozin in reversing hypoxic stress-induced cardiomyocyte damage and related cellular processes can be expected.

## 2. Results

### 2.1. NGS Biological Process, Cellular Component and Molecular Function GO Annotations

The results of the NGS RNA sequencing of rat AMI-aorta tissue showed the mechanisms that participated in AMI-mediated signaling transduction. The signaling regarding extracellular matrix structural constituents, glycosaminoglycan binding, cell adhesion molecule binding, integrin binding, heparin binding, sulfur compound binding, fibronectin binding and proteoglycan binding were affected the most in the early stage of AMI in the mouse model (Figure 1A). Figure 1B shows a volcano plot of the molecules’ different expression genes (DEGs). A volcano plot was used to display the data differences between groups from a global perspective. Most of the gene expression was not modified or modified to a small level, and only some of the genes were affected and modified significantly. In our case, there were 1113 DEGs modified in AMI-induced cardiac tissue. Most of them were activated and upregulated (1005 DEGs), and few were inactivated and downregulated (108 DEGs). The signaling transduction related to cardiomyocyte apoptosis in AMI mice was analyzed. The occurrence of this cell apoptosis also indicates that when AMI occurs, so does the death of cardiomyocytes and damage to heart tissue. Because of the heart’s very low self-repair rate, the hyperplastic connective tissue forms a scar to replace the original cardiomyocytes after damage, causing the heart to lose its initial contraction ability, and eventually lead to death (Figure 1C). We found that when AMI occurs, the signaling transduction of oxidative stress and ER stress, which are closely related to cell death, was activated in AMI-induced cardiac tissue abnormality. We found that there were 33 factors related to oxidative stress (Red). When the oxidative stress in the cardiomyocytes increased, it caused subsequent DNA damage and eventually apoptosis. Likewise, we also investigated the regulation of the cell self-protection mechanism of autophagy, and over 50 genes were regulated when AMI occurred (Green). This shows that autophagy initiated when cardiomyocytes were under pressure of AMI, but the effect on cardiomyocytes needs further experiments for interpretation (Figure 1D).

### 2.2. The Toxicity Function Analysis of RNA-seq Associated with Cardiac Tissue Remodeling, and SIRTs Involvement

RNA-sequencing analysis accompanied by IPA analysis revealed the regulation and involvement of molecules affected during AMI induction in animal models related to tissue and function (Figure 2A). It was found that AMI upregulated most of the molecular gene expression and distributed in different catalogues related to cardiac function. Heatmaps showed that the DEG catalogues associated with cardiac tissue abnormalities such as cardiac hypertrophy, cardiac dysfunction, cardiac fibrosis, cardiac necrosis/cell death, cardiac injury and cardiac damage focused on gene clusters in AMI-induced cardiac tissue alterations (Figure 2B–G). The list of genes upregulated by AMI is shown in Table 1. Among them, we found that SIRTs-related pathways play an important role in AMI-induced cardiomyopathy. Most of the molecules related to SIRT1/3/6 were affected, such as *UCP1/2*, *IL-10*, *MMP-2*, etc. In addition, using a thorough STRING analysis, we found that some of the molecules were involved in mechanisms related to cardiac tissue abnormalities, such as cardiac injury, dysfunction and cardiomyocyte death (peach box; Figure 2H). This result suggests that SIRTs and their downstream pathways may be affected at the onset of AMI and may influence the subsequent cardiac tissue remodeling process. However, the involvement of these affected pathways in the effects on cardiomyocytes remains to be elucidated.

### 2.3. Hypoxia-Mimic AMI Cell Model for the Investigation of SGLT Inhibitor Dapagliflozin’s Effect 

Based on the previous findings of the mouse AMI model, we found that many regulatory mechanisms involved in the mechanism of AMI-induced cardiomyocyte death may be affected by SGLT2 inhibitor (SGLT2i). Therefore, in the current study, we investigated the underlying mechanisms of SGLT2i for reversing AMI-induced damage to cardiomyocytes. First, an in vitro model of cardiomyocytes during AMI was simulated by culturing cells in hypoxia. Next, a CCK-8 cell viability assay was used to detect the cell proliferation of cardiomyocytes under the hypoxia model, so as to investigate the effect of hypoxia on myocardial growth and inhibition. SGLT2i dapagliflozin was additionally added to investigate the effect of reversing hypoxic stress on cell growth. Figure 3A shows that under normoxic conditions, ventricular cardiomyocyte H9c2, even in the presence of SGLT2i dapagliflozin, did not exhibit an obvious modification on affecting cell proliferation. However, after 48 h and 72 h of a hypoxia culture, a significant decrease in cell viability was observed. Interestingly, the quantitative results showed that 2.5 μM and 5 μM of dapagliflozin treatment could effectively reverse the decrease of cell viability caused by hypoxia (Figure 3B). Through flow cytometric analysis of Annexin V and 7-aminoactinomycin D (7-AAD) double staining to determine cell death, we confirmed that hypoxic stress caused increasing H9c2 cell death, including primary and secondary cell apoptosis. However, increasing the concentration of dapagliflozin reduced initial and secondary cell death by approximately 8% and 29%, respectively (Figure 3C).

### 2.4. Regulation of SIRT Family Genes by Dose-Dependent Treatment of Dapagliflozin

We found a significant change of the SIRT family in the results of the mouse cardiomyocyte with AMI model, especially a decrease of SIRT1. We believe that the SIRT family proteins may play an important role in the cardiomyocyte defect caused by AMI. Therefore, we determined the stress response of the SIRT family to hypoxia stimulation and dapagliflozin treatment in the H9c2 with hypoxia mode, including the gene expression of SIRT1/3/6. As shown in Figure 4A–C, the qPCR of SIRT1/3/6 gene expression showed that these genes were all decreased by hypoxic incubation in H9c2 cells (48 h and 72 h); however, the expression of the SIRT family was significantly increased in a dose-dependent manner after the dapagliflozin treatment. However, the increase in the 72 h hypoxia group was lower than that in the 48 h group. In the results of the gene analysis, we found that hypoxia inhibited the expression of the SIRT family genes in H9c2 cells, and dapagliflozin reversed that inhibition. Therefore, we then investigated the modification of protein expression levels of the SIRT family after dapagliflozin treatment. Using Western blot, we found that the protein expression levels of SIRT1 and SIRT3 in H9c2 were significantly decreased under hypoxia, which were consistent with the expression of the gene. Furthermore, significant increases of SIRT1 and SIRT3 protein expression were found after 48 h of dapagliflozin treatment (Figure 4D). Unfortunately, the same increase was not observed after 72 h (Figure 4E,F). Immunofluorescence staining of SIRT1 was then used to investigate the changes of SIRT1 proteins under hypoxia and dapagliflozin treatment by analyzing the modification of fluorescence intensity and expression location. As shown in Figure 4G, immunofluorescence images showed that the expression of SIRT1 in the nucleus and cytoplasm of SIRT1 decreased after 48 h of hypoxia. However, when H9c2 was treated with an additional 2.5 μM and 5 μM of dapagliflozin, it was found that the fluorescence expression of the SIRT1 protein was significantly increased, especially in the cytoplasm of SIRT1, which was stronger than that of the normal control group. Increased SIRT1 in cells may increase the deacetylation function of target proteins.

### 2.5. Next-Generation RNA Sequencing Investigation of Dapagliflozin-Induced and Reversed Hypoxia-Initiated H9c2 Cellular Mechanisms

Previous evidence has suggested the regulation of dapagliflozin in gene and protein expression of the SIRT family. In order to further understand the mechanism of hypoxia and dapagliflozin treatment in H9c2, next-generation RNA sequencing was conducted to comprehensively analyze the mechanisms of dapagliflozin’s effect on ventricular cardiomyocyte H9c2 cells and reveal the underlying protection mechanisms. Volcano plots of Figure 5A,B show the increased and decreased DEGs affected by low (2.4 µg/mL) and high (5 µg/mL) concentrations of dapagliflozin in H9c2. They reveal that the increased molecules participated when dapagliflozin altered the hypoxia-induced cellular stress in H9c2 cells. The differential expression of genes associated with the SIRT family was further demonstrated by heatmap analysis. As shown in Figure 5C, when hypoxic H9c2 cells were treated with low and high doses of dapagliflozin, significantly different DEGs were involved in gene changes in the SIRT-related pathway. In particular, the qPCR results showed that dapagliflozin could restore or upregulate the gene expression of the SIRT family-regulated molecules, including *MYCN*, *BCL2L11*, *DOLT1*, *UCP2*, *FOXO4*, *SUV39H1*, *AGTRAP*, *ESSRA*, *EPAS1* and *TP53*, which have been downregulated by hypoxic incubation (Figure 5D). Similar results were found in apoptosis-related pathways, where hypoxia induced apoptosis genes, most of which were reduced by high-concentration (10 µg/mL) dapagliflozin treatment (Figure 5E). The qPCR results also suggest that dapagliflozin restored apoptosis-related molecules such as *Bax*, *Casp8*, *NFKB1*, *NFKB2* and *RAP1A*, which were upregulated by the hypoxia culture (Figure 5F). Regarding the apoptosis pathway, we found that dapagliflozin had little effect on the related molecular genes. The main possible reason is that most of the changes in apoptotic factors were at the protein structural level, but we still saw that dapagliflozin downregulated Bcl-2 and Bcl-XL, which in turn reduced caspase 3/8/9-induced apoptosis in cardiac myocytes.

### 2.6. SIRT1 siRNA Reversed the Beneficial Effect of Dapagliflozin on H9c2 Against Hypoxic Stress

Through RNA-seq analysis, we revealed the general pathways initiated by dapagliflozin in H9c2 cardiomyocytes and the important roles played by the SIRT family. To further verify the effect of dapagliflozin on cardiomyocytes and the importance of the SIRT family, we performed a siRNA knockdown of SIRT1. First, a CCK-8 cell viability assay was conducted to investigate the effect of SIRT1 siRNA in H9c2 cell viability (Figure 6A). siSIRT1 reduced the viability of H9c2 cells by approximately 27%, suggesting that the SIRT1-related pathway plays a role in supporting cardiomyocyte survival under the stress of hypoxia. We then determined the changes in the intracellular antioxidants SOD1, SOD2 and MYC proteins, which were found to be downstream regulated by SIRT1 as a result of RNA-seq. The Western blot results showed that the expression levels of SOD2 and MYC were significantly reduced in H9c2 under hypoxic culture conditions. However, after treatment with increasing concentrations of dapagliflozin, the protein expression of SOD2 and MYC increased, whereas the expression of SOD1 did not change significantly (Figure 6B). When SIRT1 siRNA was added, we found that siSIRT1 affected the modification in SOD2 and MYC proteins. Combined with this result, it suggests that dapagliflozin may improve cell survival by initiating downstream signaling of SIRT1 such as MYC. In addition, the increase in SOD2 improves the antioxidant capacity of H9c2 cells, which is beneficial for cell survival (Figure 6C). A Kyoto Encyclopedia of Genes and Genomes (KEGG) analysis revealed the underlying mechanisms of hypoxic H9c2 and dapagliflozin stimulation. Figure 6D shows the molecular interaction networks and specific gene alterations that dapagliflozin affected and hypoxia-induced DEGs involving pathways such as SIRT and apoptosis; dapagliflozin significantly activates MYC/MYCN and cascade pathways and promotes cardiomyocyte proliferation, thereby providing cardiomyocytes with the ability to resist hypoxia.

## 3. Discussion

AMI causes severe hypoxia and ischemia in myocardial tissue, ultimately leading to cardiomyocyte apoptosis and necrosis. However, whether SGLT2i also has substantial benefits in AMI is still under investigation. The aim of this study was to investigate the protective effect of SGLT2i on cardiomyocytes in AMI. First, using the mouse AMI model, we found that the downregulation of SIRT family proteins may be involved in the process of cardiomyocyte necrosis, and KEGG analysis showed that the SIRT family may be one of the target regulatory proteins of SGLT2i dapagliflozin. Results from in vitro cell experiments showed that the protective effect of SGLT2i on hypoxic cardiomyocytes may be through the upregulation of SIR1/3/6-related signaling. Among these, dapagliflozin may activate the pro-survival mechanism of SOD2 and MYC through SIRT1 to provide cardiomyocytes with the ability to resist hypoxia. The potential signaling pathways and involvement of the SIRT family in anti-AMI were verified, providing additional evidence for the mechanism of SGLT2i in the treatment of cardiovascular disease.

There is currently little experimental evidence regarding the level and exact function of the SIRT family in cardiomyocytes after AMI. The research results of Emrullah et al. showed that the serum sirtuin 1, 3 and 6 levels in AMI patients are similar to those in normal patients, which cannot represent evidence that sirtuin 1, 3 and 6 may have a protective effect on AMI patients [26]. The clinical study results of Habieb et al. point out that the prevalence of SIRT1 carrying the rs7069102 GG genotype and G allele in MI patients was significantly higher. The HbA1c levels of MB, FBG and 2 h PPG, and total cholesterol, LDLc and CK- are significantly increased in dominant (CG + GG and CC) patients. These results suggest an increased risk of myocardial infarction in patients carrying the variant G allele of SIRT1 rs7069102 with abnormal SIRT1 function [30].

Our in vivo and in vitro results confirmed the intracellular mechanism by which AMI leads to reduction of the SIRT family and adverse cardiomyocyte survival. RNA sequencing further demonstrated that in addition to apoptosis, molecular genes involved in mitochondrial dysfunction were upregulated and that dapagliflozin could effectively reverse hypoxic stress-induced death. The data confirmed that dapagliflozin improved the downstream mechanisms of SIRT1, SIRT3, SIRT6 and MYCN, BCL2L11, DOLT1, UCP2, FOXO4, SUV39H1, AGTRAP, ESSRA, EPAS1 and TP53, which are downregulated in hypoxia. This results in improved cell proliferation and inhibition of mechanisms leading to intracellular apoptosis and oxidative stress such as Bax, Casp8, NFKB1, NFKB2 and RAP1A.

The latest study by Chen et al. indicates that the levels of SIRT1, NF-κB and CD40L in the plasma of AMI patients are significantly increased, and concludes that combined use with TNT can improve the diagnostic accuracy of AMI [29]. However, the expression and detailed role of SIRT family proteins in cardiac tissues has not been discussed. Our results suggest that SIRT1 expression is reduced in a mouse model of AMI and in hypoxia-cultured cardiomyocytes. Unexpectedly, dapagliflozin treatment increased the expression of SIRT1, 3 and 6 in the cells. We believe that SIRT1 plays an important role in regulating downstream mechanisms such as dapagliflozin upregulation of ESRRA in cardiomyocytes, which has been reported to protect cells from harmful inflammation and mitochondrial dysfunction through the activation of autophagy in a report by Kim et al. The SIRT1 siRNA results also confirmed our hypothesis that SIRT1 deficiency in H9c2 impairs the protective effect of dapagliflozin on H9c2. We have shown that dapagliflozin induces the expression of antioxidant SOD2 and promotes MYC-related cell survival mechanisms, all of which are positively regulated by SIRT1. Once SIRT1 expression is inhibited, cardiomyocyte survival is compromised. Reduced antioxidants also affect the antioxidant capacity of cardiomyocytes, further reducing cell survival. MYC is an oncogene that is highly expressed in cancer cells. There is considerable evidence in the literature that SIRT1 controls MYC-related signaling, such as ERK and MKP3 proteins, which regulate specific cell adhesion programs and promote cell growth and survival. However, we are not currently able to confirm whether the acetylation of K residues promotes cardiomyocyte survival, as the mechanisms may not be the same in cancer cells and cardiomyocytes [31,32]. Nonetheless, it is clear that dapagliflozin facilitates MYC-related signaling via SIRT1, thereby promoting the survival of cardiomyocytes under the stress of hypoxia.

In summary, the mechanism of SGLT2i dapagliflozin in treating myocardial cell injury caused by hypoxia has gradually become clear. However, the development after an AMI may be caused by multiple factors such as heart failure and other cardiovascular diseases. We believe that the SIRT family plays an important role in Dapa-induced cardiomyocyte protection. It confirms that SGLT2i can increase SIRT family proteins and help ventricular cardiomyocytes reverse damage caused by hypoxic stress. Further experiments are urgently needed to confirm its pathway and mechanism before it can be used in clinical treatment.

## 4. Materials and Methods

### 4.1. Acute Myocardial Infarction (AMI) Mouse Model

The mice were anesthetized by an intraperitoneal injection of 3% sodium pentobarbital 80 mg/kg, and the shaving surgical area was disinfected with iodine and 75% ethanol. A tracheal tube was inserted into the trachea along the glottis, the mouse was removed and connected to the ventilator. Under the armpit of the left forelimb, micro-scissors were used to open the chest between the third and fourth ribs to fully expose the heart. A small amount of the pericardium was gently picked up with micro-straight forceps, and a small amount of pericardium under the left atrial appendage was torn open to fully expose the left anterior descending coronary artery (LAD). Using a 7-0 suture with a needle, the needle was inserted 2 mm from the lower edge of the left atrial appendage, and the suture was passed through the LAD to completely block the LAD blood flow. After the ligation was completed, 6-0 sutures were used to completely close the chest opening. Close attention was paid to the mouse’s condition after surgery, including whether there were respiratory abnormalities, etc. After the mice woke up naturally, the respirator was removed, and they rose normally.

### 4.2. Cell Culture of Human Cardiomyocyte H9c2

H9c2 cells derived from rat embryonic myoblasts (ATCC CLR-1446; Rockville, MD, USA) were cultured according to our previous report. Briefly, H9c2 cells were seeded onto collagen-coated dishes and cultured in DMEM supplemented with 10% fetal bovine serum (FBS). Culture conditions were 5% CO_2_ and 95% air, 37 °C within passages 20 to 40.

### 4.3. The Hypoxia Cell Culture

Cell culturing was done in flasks, plates or petri dishes. To open the chamber, two white plastic clips were first opened on the chamber connection tube (the tube used to inject/purge the low oxygen gas), then the steel ring clip was gently opened. The pipeline was connected to a “hypoxic tank” containing a 1% O_2_ gas mixture to produce hypoxia. The chamber was re-inflated once in 1–3 h to eliminate the O_2_ contained in the medium.

### 4.4. SGLT2 Inhibitor Dapagliflozin (DAPA) Treatment

H9c2 cells were treated with dapagliflozin at concentrations according to previously published articles. After the hypoxic culture, the cells were treated with 2.5 and 5 μM for 48 h for cell morphology observation. The cell supernatant was removed, and the cells were harvested after washing twice with PBS. RNA extraction kit Trizol (1 mL; Thermo Fisher Scientific, Carlsbad, CA, USA) was used to collect cellular mRNA for cDNA reverse transcription or use a protein lysis buffer to collect the extracted protein and perform protein quantification before subsequent experiments.

### 4.5. Cell Survival Assay

The survival of H9c2 cells was determined by hypoxic incubation and treatment with the indicated concentrations of dapagliflozin using the CCK-8 cell proliferation assay kit according to the manufacturer’s instructions (Thermo Fisher Scientific, Carlsbad, CA, USA). Briefly, 1 × 10^5^ cells were seeded and treated with the indicated concentrations of dapagliflozin (Sigma-Aldrich, St. Louis, MO, USA) and/or hypoxic incubation (1% O_2_) for 48 and 72 h. After incubation, 10 µL of 12 mM CCK-8 stock solution was added and incubated at 37 °C for 4 h, followed by the addition of 50 µL of DMSO, which was mixed and the absorbance measured at 540 nm. 

### 4.6. Flow Cytometry Analysis

Flow cytometry-based cell cycle assays were performed as described in the instruction manual. The cells were then rinsed twice with pre-cooled PBS and collected by centrifugation at 200 g for 5 min at 4 °C. After centrifugation, the cells were resuspended in 1 mL of 7-AAD staining buffer (10 µg/mL RNase A, 50 µg/mL PI, PBS) and double-stained with an Annexin V staining kit (PharMingen, San Diego, CA, USA) and incubated at 37 °C for 30 min. The cells were then analyzed using a FACSCalibur flow cytometer (Becton Dickinson, Mountain View, CA, USA), and the results were analyzed using WinMDI 2.8 software (written by Joseph Trotter, Scripps Research Institute, La Jolla, CA, USA).

### 4.7. mRNA Collection and qRT-PCR Analysis

A Nucleospin RNA kit was used (Macherey-Nagel, Allentown, PA, USA) to isolate total RNA. Cell lysates were applied to RNA columns and centrifuged at 15,000 RPM. A membrane desalting buffer was used and added to the column, followed by centrifugation. An RNase-free DNase reaction mixture was used to ablate the DNA in each column. After the DNAse reaction was complete, the column containing the RNA was transferred to a new tube for final RNA sample purification. Finally, the RNA sample was dissolved using RNase-free HO and transfered to a new 1.5 mL tube. All RNA samples were stored at −80 °C before complementary DNA (cDNA) was synthesized using reverse transcription reagents (Applied Biosystem, Waltham, MA, USA). The prepared cDNA was then left in a 96-well quantitative polymerase chain reaction (qPCR) for the analysis of gene expression.

### 4.8. Protein Preparation and Western Blotting

The protein lysates were separated according to gel chemistry using SDS-polyacrylamide gel electrophoresis (SDS-PAGE) for wet transfer, and a transfer buffer was prepared. PVDF transfer membranes were prepared, and the manufacturer’s instructions were followed for dry film preparations by prewetting in ethanol (100%) for 30 s, rinsing briefly in deionized water, and equilibrating in a transfer buffer for 5 min. The membrane was incubated with a sufficient volume of a blocking buffer for 30–60 min at room temperature with agitation. After diluting the primary antibody in a blocking buffer, it reacted overnight and was then incubated with the corresponding secondary antibody. Chemiluminescence was then excited by ELC, and the expression of the target protein was observed.

### 4.9. Immunofluorescence Staining and Microscopy Analysis

To analyze cardiomyocyte cells using immunofluorescence assays, as suggested by the user manual, 1:100–1:200 dilutions of the target primary antibody were prepared and reacted with the antigen in the cells. After washing, slides were incubated with DAPI-conjugated antibodies bound to antibodies of different conditions and subsequently visualized under a fluorescence microscope (Olympus BX51, Olympus, Tokyo, Japan).

### 4.10. Biological Pathway Analysis

Alignments of high-quality reads were aligned to the human reference genome (grch38.p7) and imported into Ingenuity’s IPA 8.7 software. After calculating the participation of featureCounts for each mechanism, the performance was quantified with RLE/TMM/FPKM. Differentially expressed genes were screened with the q value <0.05 as the screening threshold. The Ingenuity knowledge base used in this analysis consisted of genes only, and both direct and indirect relationships were considered. The Diseases and Disorders, Molecular and Cellular Functions, Physiological Phylogenetics and Functions, and Top Canonical Pathways results were associated with a given dataset at *p* < 0.05.

### 4.11. Lentiviral Vector Delivery of SIRT1 siRNA Encoding

H9c2 cells at 50% confluence were used for lentivirus transfection. A mixture of 100 μL SIRT1 lentiviral plasmid (plasmid ID: TRCN0000306512) and 1 mL H-DMEM (supplemented with 5% FBS, 5% BCS, 1% antibiotic–antifungal solution and 1% L-glutamine solution) was added to H9c2 cells in a 6-well plate. The cells were incubated in a 37 °C, 5% CO_2_ incubator for 24 h of infection. A medium containing 25 mg/mL puromycin was added for 24 h for cell selection. Cells with resistance to puromycin were amplified and cultured, and the knockdown efficiency was confirmed by Western blot analysis.

### 4.12. Statistics

All data were expressed as the mean ± standard error of the mean (SEM). SPSS 13.0 (SPSS13.0 Software Inc., Armonk, New York, NY, USA) was used for one-way analysis of variance differences between groups. Comparisons between groups were conducted through Duncan’s multi-range test. *p* < 0.05 was considered statistically significant.

## Figures and Tables

**Figure 1 ijms-25-08541-f001:**
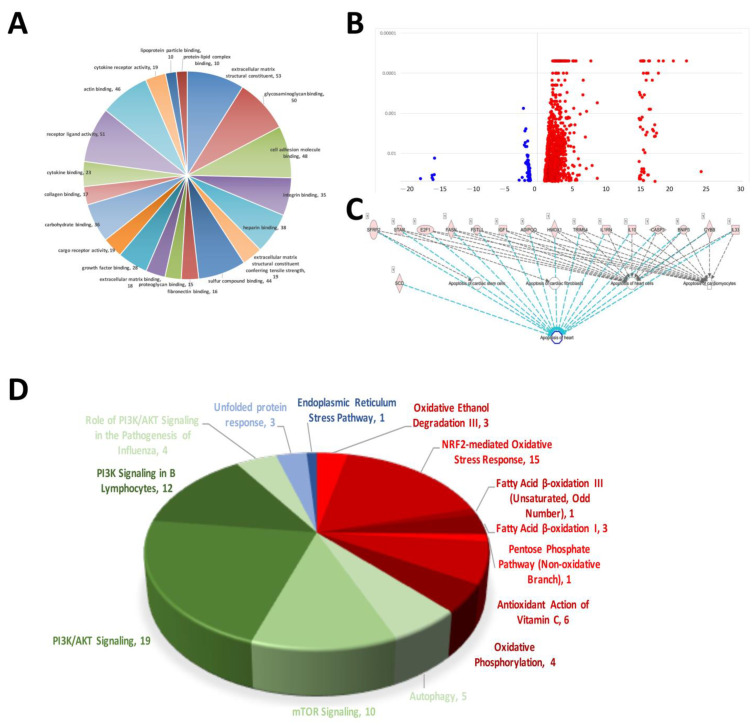
Results of myocardial RNA-sequencing analysis of AMI mice. (**A**) Pie chart of mRNA alterations in cellular process mechanisms in cardiac tissue during mouse AMI. (**B**) Volcano plot of RNA sequencing showing increased and decreased differentially expressed genes (DEGs). (**C**) Molecular prediction of sequenced DEGs associated with apoptosis. (**D**) DEGs involved in intracellular stress pathways and signaling that are associated with the development of cardiovascular disease.

**Figure 2 ijms-25-08541-f002:**
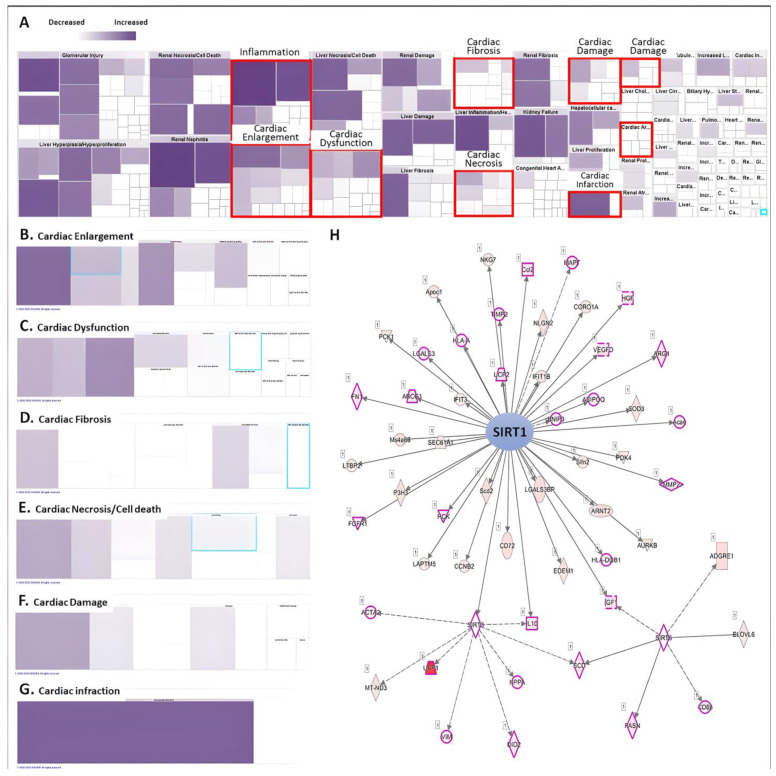
Analysis of RNA-seq using Ingenuity Pathway Analysis (IPA). (**A**) Heatmap analysis of disease and biological function of DEGs. The level of purple represents the number of genes involved. The subcategory analyzes the risk of heart disease and dysfunction caused by AMI, including (**B**) cardiac enlargement, (**C**) cardiac dysfunction, (**D**) cardiac fibrosis, (**E**) cardiac necrosis and cell death, (**F**) cardiac damage and (**G**) cardiac infraction. (**H**) IPA analysis predicts key molecules involved in upstream regulation in mice with myocardial AMI.

**Figure 3 ijms-25-08541-f003:**
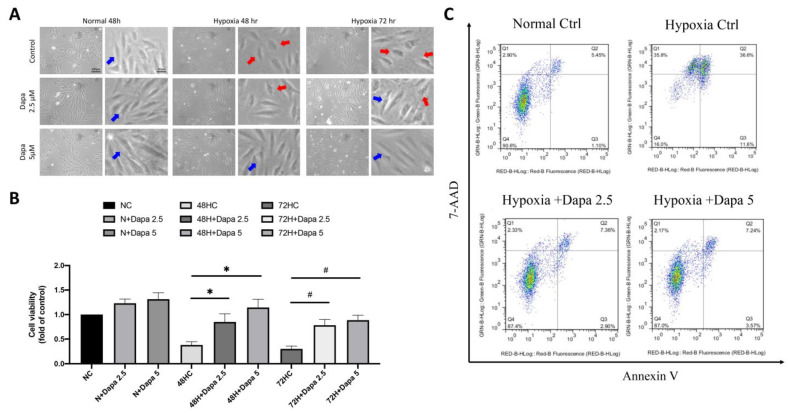
SGLT2 inhibitors reversed ventricular cardiomyocyte death caused by hypoxia. (**A**) Cell appearance of H9c2 and DAPA-treated cardiac ventricular cells cultured in hypoxic mode. Blue arrow: health H9c2; Red arrow: hypoxic injury H9c2. Magnification:100× and 200×. (**B**) Cell proliferation assay of H9c2 in hypoxic incubation and DPA treatment in time- and dose-dependent manner. (**C**) Flow cytometry analysis of Annexin V and 7-aminoactinomycin D (7-AAD) assay for primary and secondary cell apoptosis investigation. * *p* < 0.05 compared with 48 hr hypoxia control cell (48HC); # *p* < 0.05 compared with 72 hr hypoxia control cell (72HC).

**Figure 4 ijms-25-08541-f004:**
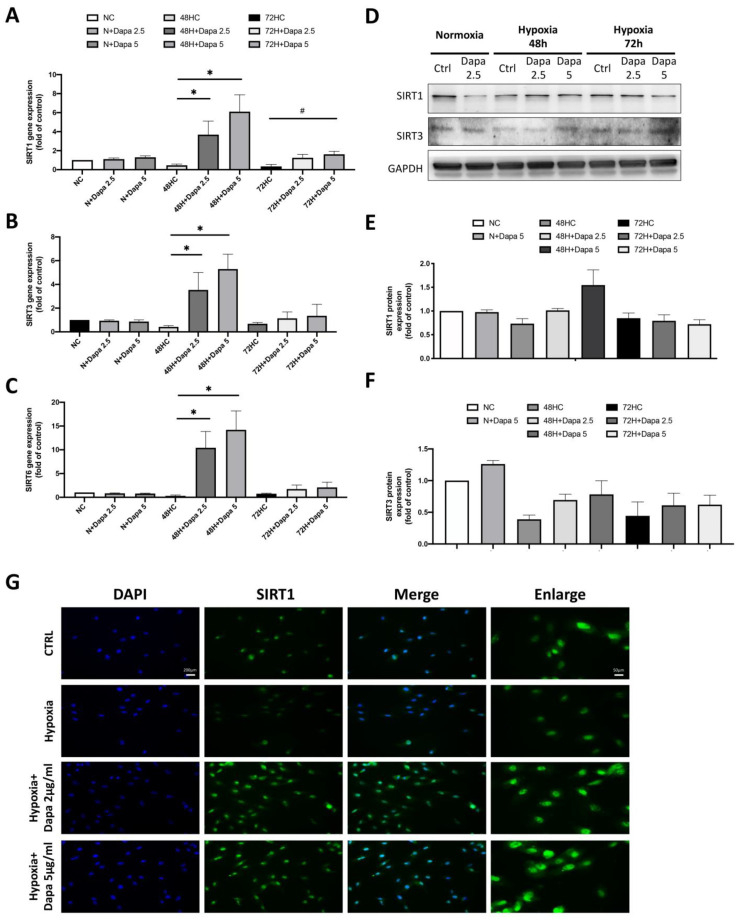
Changes in the expression of the SIRT family in cardiomyocyte H9c2 induced by hypoxic culture and dapagliflozin treatment. qPCR analysis of gene expression after 48 and 72 h of hypoxia and dapagliflozin showing changes in the SIRT family, including (**A**) SIRT1, (**B**) SIRT3 and (**C**) SIRT6. (**D**) SIRT1 and SIRT3 protein changes after hypoxia and dapagliflozin treatment. Quantitative and statistical analysis of (**E**) SIRT1 and (**F**) SIRT3 protein expression regulation. (**G**) Fluorescent blue analysis of the positional differences in the expression of SIRT1 in cells after hypoxia and dapagliflozin treatment. Magnification:100× and 200×. * *p* < 0.05 compared with 48HC; # *p* < 0.05 compared with 72HC.

**Figure 5 ijms-25-08541-f005:**
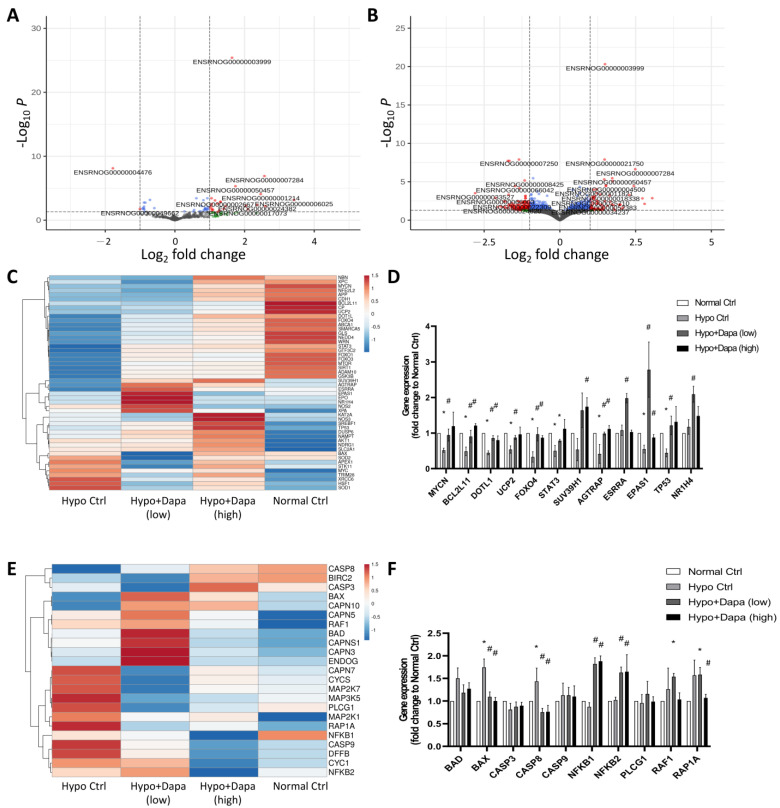
Next-generation RNA sequencing was used to analyze the effect of dapagliflozin on rescuing hypoxic cardiomyocyte H9c2. RNA-sequencing volcano plots of hypoxia-cultured H9c2 at (**A**) low (2.4 µg/mL) and (**B**) high (5 µg/mL) concentrations of dapagliflozin treatment. (**C**) Heatmap analysis of the trends in gene changes related to SIRTs and (**D**) verification qPCR analysis. (**E**) Heatmap of DEGs involved in apoptosis-associated signaling, and (**F**) qPCR verification in hypoxic H9c2 cells incubated with dapagliflozin. * *p* < 0.05 compared with normal control; # *p* < 0.05 compared with hypoxia control.

**Figure 6 ijms-25-08541-f006:**
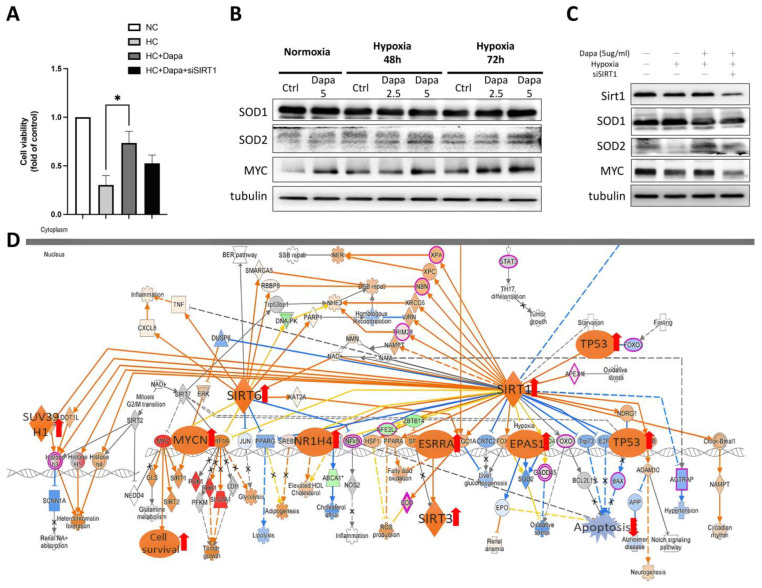
The effect of SIRT1 siRNA on reversing dapagliflozin’s beneficial effect on hypoxic cardiomyocytes. (**A**) Cell viability of SIRT1 siRNA on hypoxic H9c2. (**B**) Western blot of antioxidants SOD1, SOD2 and pro-cell survival regulator MYC, which were affected by hypoxia and dapagliflozin treatment. (**C**) The modification of SIRT1 downstream factors protein expression regulated by SIRT siRNA. (**D**) KEGG analysis of SIRT-associated signaling showed that DAPA-affected DEGs were involved in H9c2. * *p* < 0.05 compared with indicated hypoxia H9c2 control (HC).

**Table 1 ijms-25-08541-t001:** List of candidate genes that may be involved in AMI-caused cardiac myocardial pathology and remodeling.

Categories	Diseases or Functions Annotation	Molecules Counts	Molecules	*p*-Value
**Cardiac Enlargement**	Enlargement of heart	61	ACTG2,ADA,ADIPOQ,ALDH1A2,APOE,BGN,BNIP3,CACNA1H,CACNB3,Ccl2,COX7A1,CPT2,CTSC,CYBB,EGFR,EGLN3,ELN,FASN,FGFR1,GATA5,GJA5,GPX1,GPX3,GPX7,GUCY1A1,HCK,HMOX1,HOPX,IGF1,IL17RA,IL33,ITGAV,ITGB3,LIF,MDK,MMP2,MYOZ2,NCF1,NPPA,P2RY6,PDGFC,PITX2,PLA2G4A,PLIN5,POSTN,PRKCD,PTGS2,S100A10,S100A6,SDC1,SERPINE1,SLC1A3,SLCO2A1,STAB1,TCF15,THBS2,THBS4,TIMP1,TNFRSF11B,TNFRSF1B,TRIM54	1.99 × 10^−8^
	Hypertrophy of heart	42	ACTG2,ADA,ADIPOQ,APOE,CACNA1H,Ccl2,CPT2,CTSC,CYBB,EGFR,ELN,FASN,GATA5,GPX3,GUCY1A1,HCK,HMOX1,HOPX,IGF1,IL33,ITGB3,LIF,MDK,MYOZ2,NCF1,NPPA,P2RY6,PDGFC,PLA2G4A,PLIN5,POSTN,PRKCD,PTGS2,S100A10,S100A6,SERPINE1,TCF15,THBS4,TIMP1,TNFRSF11B,TNFRSF1B,TRIM54	8.04 × 10^−6^
**Cardiac Dysfunction**	Left ventricular dysfunction	25	ADIPOQ,APOE,BGN,BNIP3,Ccl2,CCR2,CYBB,GATA5,GPX1,HMOX1,IL10,ITGB3,MDK,MGP,MMP2,NCF1,NPPA,PLIN5,POSTN,PTX3,SERPINE1,SLCO2A1,SPP1,TIMP1,TLR2	5.4 × 10^−6^
	Dysfunction of heart	26	ADIPOQ,APOE,BGN,BNIP3,Ccl2,CCR2,CYBB,GATA5,GPX1,HMOX1,IL10,ITGB3,MDK,MGP,MMP2,NCF1,NPPA,PLIN5,POSTN,PTX3,SCD,SERPINE1,SLCO2A1,SPP1,TIMP1,TLR2	1.48 × 10^−4^
**Cardiac Fibrosis**	Fibrosis of heart	26	APOE,BGN,BUB1B,CACNA1H,CDON,CYBB,EGR1,GPX1,HMOX1,HOPX,IL10,KRT18,NCF1,NPPA,PADI4,PDGFC,POSTN,PTGS2,PTX3,SERPINE1,SGCA,SPP1,THBS2,TIMP1,TLR2,TNFRSF1B	5.2 × 10^−4^
	Interstitial fibrosis of left ventricle	4	CYBB,GPX1,NPPA,PTX3	1.46 × 10^−3^
	Interstitial fibrosis of heart	7	BUB1B,CYBB,GPX1,NCF1,NPPA,POSTN,PTX3	4.89 × 10^−3^
**Cardiac Necrosis/Cell Death**	Cell death of heart	22	ADIPOQ,BNIP3,CASP3,CCN4,CYBB,E2F1,FASN,FSTL1,HMOX1,IGF1,IL10,IL1RN,IL33,PADI4,SCD,SFRP2,SGCA,SPRR1A,STAR,THBS2,THBS4,TRIM54	8.84 × 10^−5^
	Cell death of heart cells	18	ADIPOQ,BNIP3,CASP3,CCN4,CYBB,E2F1,FASN,FSTL1,HMOX1,IGF1,IL10,IL1RN,IL33,SFRP2,SPRR1A,STAR,THBS2,TRIM54	1.2 × 10^−3^
	Apoptosis of heart	16	ADIPOQ,BNIP3,CASP3,CYBB,E2F1,FASN,FSTL1,HMOX1,IGF1,IL10,IL1RN,IL33,SCD,SFRP2,STAR,TRIM54	2.44 × 10^−3^
	Necrosis of cardiac muscle	17	ADIPOQ,BNIP3,CASP3,CCN4,CYBB,E2F1,FASN,FSTL1,HMOX1,IGF1,IL10,IL1RN,IL33,SGCA,SPRR1A,THBS2,TRIM54	3.45 × 10^−3^
	Apoptosis of heart cells	15	ADIPOQ,BNIP3,CASP3,CYBB,E2F1,FASN,FSTL1,HMOX1,IGF1,IL10,IL1RN,IL33,SFRP2,STAR,TRIM54	4.2 × 10^−3^
	Cell death of cardiomyocytes	16	ADIPOQ,BNIP3,CASP3,CCN4,CYBB,E2F1,FASN,FSTL1,HMOX1,IGF1,IL10,IL1RN,IL33,SPRR1A,THBS2,TRIM54	5.05 × 10^−3^
	Apoptosis of cardiomyocytes	13	ADIPOQ,BNIP3,CASP3,CYBB,E2F1,FASN,FSTL1,HMOX1,IGF1,IL10,IL1RN,IL33,TRIM54	0.0168
**Cardiac Damage**	Rupture of heart	5	BGN,GDF15,POSTN,SERPINE1,TRIM54	5.22 × 10^−5^
	Damage of heart	10	ADIPOQ,BGN,GDF15,HMOX1,IL10,INHBA,POSTN,SERPINE1,THBS4,TRIM54	2.76 × 10^−3^
	Rupture of heart ventricle	2	BGN,POSTN	3.02 × 10^−3^
	Rupture of myocardium	2	GDF15,SERPINE1	8.74 × 10^−3^
	Damage of myocardium	3	ADIPOQ,GDF15,SERPINE1	0.0149
	Rupture of left ventricle	1	BGN	0.055
	Reperfusion injury of myocardium	1	ADIPOQ	0.055
	Cardiotoxicity	1	CYBB	0.156
	Injury of heart	2	ADIPOQ,INHBA	0.442
**Cardiac Infarction**	Myocardial infarction	22	ACTA2,ADIPOQ,APOE,Ccl2,Ccl7,CD44,COL3A1,CSF1R,CSF3R,CXCR4,CYBB,FCER1G,FSTL1,GDF15,HMOX1,IL10,KRT18,KRT8,NCF2,OLR1,PADI4,TNFRSF1B	2.89 × 10^−9^
	Early stage myocardial infarction	2	Ccl2,Ccl7	8.74 × 10^−3^

## Data Availability

Data are contained within the article.

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
