# Peer review of "The Beneficial Effect of the SGLT2 Inhibitor Dapagliflozin in Alleviating Acute Myocardial Infarction-Induced Cardiomyocyte Injury by Increasing the Sirtuin Family SIRT1/SIRT3 and Cascade Signaling"

_ijms, 2024, doi:10.3390/ijms25158541_

Round 1

Reviewer 1 Report

Comments and Suggestions for Authors

The authors report that SGLT2i improve cardiac injury during MI, and that this benefit is mediated via SIRT1 and SIRT3

The authors are to be praised for the novelty of their findings; most research on SGLT2i focuses on heart failure, but the authors focus on ischemia-reperfusion which is much less studied. Second, the focus on //R injury is very timely following the very recent publication of the EMPACT-MI and DAPA-MI trials about SGLT2i during MI. Third, the methods are solid, the study is done at two levels (cardiomyocyte in vitro and also mice model of coronary ligation), the results are concordant, the authors report NGS (not usually utilized for SLGT2i), the conclusions are supported by the data, and the manuscript reads well.

This reviewer only identifies the following issues:

            Major issues:

-       EMPACT-MI has already been published (please quote N Engl J Med. 2024 Apr 25;390(16):1455-1466.). The authors should explain that empagliflozin initiated within 14 days after MI does not improve outcomes of CV ceath+HF hospitalization

-       DAPA-MI And EMPACTMI show that SGLT2i after MI do not improve outcome. However, the main message is that SGLT2i initiated before MI ameliorate myocardial ischemia-reperfusion injury and reduce MI size, namely chronic treatment with SGLT2i before ACS reduce MI size when it might happen. In fact, the authors should mention that 1) SGLT2i treatment ameliorates myocardial ischemia-reperfusion and reduces MI size (as per LGE-MRI and TTC-histology) in porcine model of LAD occlusion (please quote Circ Cardiovasc Imaging. 2023 Apr;16(4):e015298); and 2) this is is supported by data in human T2DM patients that SGLT2i users have smaller MI size (as per peak troponin levels) than non-SGLT2i users (please quote Cardiovasc Diabetol  2022;21:77)

-       The authors analyze genes in ischemic cardiomyopathies in MI mode in mice. The authors should additionally present the gold-standard measurement of MI size as per TTC and L function (echocardiography or MRI or PV loop).

-       The authors should demonstrate in vitro whether SGLT2i act via sirtuin activation. For this, the authors should demonstrate that the benefits of SGLT2i in vitro (apoptosis, cell death, oxidative stress) are mitigated in the presence of pharmacological sirtuin inhibition

            Minor issues:

-       Introduction: When discussing EMPEROR trials, the authors should mention that SGLT2i not only improve outcomes but also quality of life (please quote Diabetes Metab Syndr. 2022 Feb;16(2):102417)

-       Introduction:  When discussing the mechanism of action of SGLT2i, the authors should highlight the three main mechanisms postulated so far:

o   SGLT2i inhibits sodium-hydrogen exchanger NHE), please quote Diabetologia. 2018 Mar;61(3):722-726;

o   SGLT2i induce a metabolic shift in cardiac fuel utilization away from the energy-inefficient glucose towards the utilization of fatty acids and ketone bodies, which enhance energetics (please quote Circulation. 2022 Sep 13;146(11):819-82);

o   heart failure is associated with functional iron deficiency due to increased plasma hepcidin levels, and lower iron content in tissues thus worsening mitochondria,. Conversely, SGLT2i improve iron metabolism by reducing hepcidin, which improves iron content in tissues and mitochondria function (please quote Nature Cardiovascular Research 2023;2:1032-1043).

Comments on the Quality of English Language

Language is OK

Author Response

Reviewer 1:

The authors report that SGLT2i improve cardiac injury during MI, and that this benefit is mediated via SIRT1 and SIRT3

The authors are to be praised for the novelty of their findings; most research on SGLT2i focuses on heart failure, but the authors focus on ischemia-reperfusion which is much less studied. Second, the focus on //R injury is very timely following the very recent publication of the EMPACT-MI and DAPA-MI trials about SGLT2i during MI. Third, the methods are solid, the study is done at two levels (cardiomyocyte in vitro and also mice model of coronary ligation), the results are concordant, the authors report NGS (not usually utilized for SLGT2i), the conclusions are supported by the data, and the manuscript reads well.

This reviewer only identifies the following issues:

Major issues:

EMPACT-MI has already been published (please quote N Engl J Med. 2024 Apr 25;390(16):1455-1466.). The authors should explain that empagliflozin initiated within 14 days after MI does not improve outcomes of CV ceath+HF hospitalization.

Response: Thanks for reviewer’s suggestion. We already revised the manuscript as reviewer’s suggestion.

Revised manuscript:

EMPACT-MI study also showed empagliflozin initiated within 14 days after MI did not decrease the CV death or heart failure hospitalization, but it could reduce the risk of HF after heart attack.

(Introduction section, Page 2, line 93-96)

DAPA-MI And EMPACTMI show that SGLT2i after MI do not improve outcome. However, the main message is that SGLT2i initiated before MI ameliorate myocardial ischemia-reperfusion injury and reduce MI size, namely chronic treatment with SGLT2i before ACS reduce MI size when it might happen. In fact,

the authors should mention that 1) SGLT2i treatment ameliorates myocardial ischemia-reperfusion and reduces MI size (as per LGE-MRI and TTC-histology) in porcine model of LAD occlusion (please quote Circ Cardiovasc Imaging. 2023 Apr;16(4):e015298); and 2) this is is supported by data in human T2DM patients that SGLT2i users have smaller MI size (as per peak troponin levels) than non-SGLT2i users (please quote Cardiovasc Diabetol 2022;21:77)

Response: Thanks for reviewer’s suggestion. We already revised the manuscript and added the references to studies demonstrating the benefits of SGLT2i in porcine models and human T2DM patients.

Revised manuscript

Recent studies have also highlighted the benefits of SGLT2i in preclinical and clinical settings beyond heart failure. A study using a porcine model of left anterior descending coronary artery occlusion found that SGLT2i ameliorates myocardial ischemia-reperfusion injury and significantly reduces MI size (18). Additionally, clinical data from T2DM patients indicate that those treated with SGLT2i have smaller MI sizes, as evidenced by lower peak troponin levels compared to non-SGLT2i users (19).

(Introduction section, Page 2, line 81-86)

-  The authors analyze genes in ischemic cardiomyopathies in MI mode in mice. The authors should additionally present the gold-standard measurement of MI size as per TTC and L function (echocardiography or MRI or PV loop).

Response: Thanks for the value comments. We don’t measure echo of the MI mouse LV function, instead, the echo of the ventricular blood flow was measured. Because the LAD was clamped with micro straight forceps, the LAD blood flow of mice in the AMI group was significantly reduced compared to the mice in the normal control group. The mice were still breathing, living and eating normally after surgery.

- The authors should demonstrate in vitro whether SGLT2i act via sirtuin activation. For this, the authors should demonstrate that the benefits of SGLT2i in vitro (apoptosis, cell death, oxidative stress) are mitigated in the presence of pharmacological sirtuin inhibition

Response: Thanks for the valuable comments. We have completed the SIRT1 knockdown experiments. Using SIRT1 siRNA, we verified that SIRT1 knockdown in cardiomyocytes would attenuate the protection against hypoxia provided by dapagliflozin. siSIRT1 would lead to endogenous SIRT1 deficiency, which would prevent the downstream proteins of SIRT1, such as SOD2, MYC, etc., from being upregulated after dapagliflozin treatment. We believe that dapagliflozin induces the expression of antioxidant SOD2 and promotes MYC-related cell survival mechanisms, all of which are positively regulated by SIRT1. Once SIRT1 expression is inhibited, cardiomyocyte survival is compromised. Reduced antioxidants also affect the antioxidant capacity of cardiomyocytes, further reducing cell survival. MYC is an oncogene that is highly expressed in cancer cells. There is considerable evidence in the literature that SIRT1 controls MYC-related signaling, such as ERK and MKP3 proteins, which regulate specific cell adhesion programs and promote cell growth and survival.

The results also further support the hypothesis of our study and clarifies the mechanism by which dapagliflozin confers resistance to hypoxia in cardiomyocytes.

(Results section, Page 12, line 439-473, Figure 6, Discussion section, Page 14, line 544-558)

Minor issues:

- Introduction: When discussing EMPEROR trials, the authors should mention that SGLT2i not only improve outcomes but also quality of life (please quote Diabetes Metab Syndr. 2022 Feb;16(2):102417)
Response: Thanks for reviewer’s suggestion. We already revised our manuscript and added the reference.

Revised manuscript

According to the previous HF studies, SGLT2i can improve not only CV out-comes, but also quality of life for HF

(Introduction section, Page 2, line 78-80)

- Introduction:  When discussing the mechanism of action of SGLT2i, the authors should highlight the three main mechanisms postulated so far:

o   SGLT2i inhibits sodium-hydrogen exchanger NHE), please quote Diabetologia. 2018 Mar;61(3):722-726;

o   SGLT2i induce a metabolic shift in cardiac fuel utilization away from the energy-inefficient glucose towards the utilization of fatty acids and ketone bodies, which enhance energetics (please quote Circulation. 2022 Sep 13;146(11):819-82);

o   heart failure is associated with functional iron deficiency due to increased plasma hepcidin levels, and lower iron content in tissues thus worsening mitochondria,. Conversely, SGLT2i improve iron metabolism by reducing hepcidin, which improves iron content in tissues and mitochondria function (please quote Nature Cardiovascular Research 2023;2:1032-1043).

Response: Thanks for reviewer’s suggestion. We already revised manuscript and highlight the three main mechanisms of SGLT2i.

Revised manuscript

Firstly, SGLT2i inhibit the sodium-hydrogen exchanger (NHE), which plays a critical role in cardiac cellular homeostasis. Inhibition of NHE by SGLT2i helps to reduce intracellular sodium and calcium overload, thereby protecting the myocardium from ischemic injury (11). Secondly, SGLT2i induce a metabolic shift in cardiac fuel utilization. This shift moves away from the energy-inefficient glucose metabolism towards the more efficient use of fatty acids and ketone bodies. This metabolic reprogramming enhances myocardial energetics and efficiency, contributing to improved cardiac function (12). Thirdly, HF is often associated with functional iron deficiency, characterized by increased plasma hepcidin levels and lower iron content in tissues, which worsens mitochondrial function. SGLT2i have been shown to improve iron metabolism by reducing hepcidin levels, thereby increasing iron content in tissues and enhancing mitochondrial function. This mechanism further supports the cardioprotective effects of SGLT2i (13).

(Introduction section, Page 2, line 58-71)

Reviewer 2 Report

Comments and Suggestions for Authors

The study by Lin et al presents important data on the the beneficial effect of the SGLT2 inhibitor dapagliflozin in alleviating acute myocardial infarction-induced cardiomyocyte  injury by increasing the Sirturin family SIRT1/SIRT3 and cascade signaling. The study is significant and novel and I must commend the authors. The only major concern I have is that the flow, presentation and grammatical errors may be overshadowing the impact. Below are my major concerns: and suggestions

1. In the introduction SGLT2I should be defined and abbreviated thereon. please do this for all abbreviations like AMI etc.

2. Grammar correction is required throughout the manuscript i.e second sentence of introduction "drugs" should be "drug"; see also sentence that begins on line 62 etc.

3. Introduction could be shortened

4. Coherency is lacking in the presentation of results. This could be because the aims are not very clearly outlined making it difficulty to follow. I would suggest that the aims and research questions are clearly outlined and the results should follow the sequence of the outlined aims to improve flow and keep the presentation more focused.

5. The figures are not legible. Kindly make them more clearly.

6. The methods could come before the results to improve coherency and presented in a manner to show sequence of events. 

Comments on the Quality of English Language

too many grammatical errors requiring correction throughout

Author Response

Reviewer 2

The study by Lin et al presents important data on the the beneficial effect of the SGLT2 inhibitor dapagliflozin in alleviating acute myocardial infarction-induced cardiomyocyte injury by increasing the Sirturin family SIRT1/SIRT3 and cascade signaling. The study is significant and novel and I must commend the authors. The only major concern I have is that the flow, presentation and grammatical errors may be overshadowing the impact. Below are my major concerns: and suggestions

  1. In the introduction SGLT2I should be defined and abbreviated thereon. please do this for all abbreviations like AMI etc.

Response: Thanks for reviewer’s comment. We already revised the manuscript as reviewer’s suggestion.

  1. Grammar correction is required throughout the manuscript i.e second sentence of introduction "drugs" should be "drug"; see also sentence that begins on line 62 etc.

Response: Thanks for reviewer’s comment. We already revised the manuscript as reviewer’s suggestion.

  1. Introduction could be shortened

Response: Thanks for reviewer’s comment. We already shortened our introduction section.

  1. Coherency is lacking in the presentation of results. This could be because the aims are not very clearly outlined making it difficulty to follow. I would suggest that the aims and research questions are clearly outlined and the results should follow the sequence of the outlined aims to improve flow and keep the presentation more focused.

Response: Thanks for your valuable comments. We have reworded the discussion part of this article, highlighting the research purpose and hypothesis first, and then discussing our research findings in order and comparing their significance. We hope that this change will make the article easier to read.

(Discussion section, Page 13-14, line 477-487, 525-534, 544-558, 601-608)

  1. The figures are not legible. Kindly make them more clearly.

Response: We have increased the resolution of the images (600 dpi) to solve the problem of blurred images.

  1. The methods could come before the results to improve coherency and presented in a manner to show sequence of events.

Response: Thank you for your valuable comments, we have placed the methodology of the study before the sentences describing the results in the article to improve consistency and also to improve the flow of reading the article.

(Results section, Page 3-12)

Round 2

Reviewer 1 Report

Comments and Suggestions for Authors

The authors have satisfactorily addressed all my previous comments